# The Multiple Dimensions of Networks in Cancer: A Perspective

Cristian Axenie [1,*,†] , Roman Bauer [2,†] and María Rodríguez Martínez [3,†]

1   Audi Konfuzius-Institut Ingolstadt Lab at the Technische Hochschule Ingolstadt, 85049 Ingolstadt, Germany
2   Department of Computer Science, University of Surrey, Guildford GU2 7XH, UK; r.bauer@surrey.ac.uk
3   IBM Research Europe, 8803 Rüschlikon, Switzerland; mrm@zurich.ibm.com
*   Correspondence: cristian.axenie@audi-konfuzius-institut-ingolstadt.de
†   All authors contributed equally to this work. Order is alphabetic.

**Abstract:** This perspective article gathers the latest developments in mathematical and computational oncology tools that exploit network approaches for the mathematical modelling, analysis, and simulation of cancer development and therapy design. It instigates the community to explore new paths and synergies under the umbrella of the Special Issue "Networks in Cancer: From Symmetry Breaking to Targeted Therapy". The focus of the perspective is to demonstrate how networks can model the physics, analyse the interactions, and predict the evolution of the multiple processes behind tumour-host encounters across multiple scales. From agent-based modelling and mechano-biology to machine learning and predictive modelling, the perspective motivates a methodology well suited to mathematical and computational oncology and suggests approaches that mark a viable path towards adoption in the clinic.

**Keywords:** mathematical and computational oncology; cancer; networks; mechano-biology; machine learning

## 1. Introduction: Networks in Cancer

Cancer is a highly complex condition that causes approximately 10 million annual deaths in the world, according to the International Agency for Research on Cancer of the World Health Organization [1]. While cancer comprises a group of heterogeneous diseases, genomic alterations result in uncontrolled cellular proliferation and progression to metastasis in most cases.

The last decades have shown that, in addition to genomic alterations, the microenvironment surrounding cancer cells also plays an essential role in shaping and enabling a whole range of underlying fundamental processes. Beyond gene expression, the microenvironment influences tumour initiation, progression, immune evasion, and treatment response. As cancer progresses from a small clone of abnormal cells to a clinically apparent disease, tumours alter the structure and function of surrounding tissues through biochemical and physical processes. Notably, these alterations comprise, among others, factors promoting vasculogenesis and angiogenesis. Overall, such processes shape the abnormal physiology of the tumour–host interaction and are often responsible for tumourigenesis and treatment resistance.

Network theory is helpful to model and investigate the abnormal molecular and cellular processes associated with cancer. Capable of capturing dynamics and interactions at very different spatial and temporal scales, e.g., gene regulatory networks, protein–protein interaction networks, cellular communication events, etc., network theory provides a very attractive modelling and computational approach for understanding cancer complexity. Interestingly, the tools and approaches used in the field of mathematical and computational oncology have very often been used in other disciplines, such as computational neuroscience, ecology or artificial intelligence, the overlapping point being the use of modelling techniques based on the analysis of complex networks.

The current review attempts to unify different perspectives regarding the use of network theory to advance cancer research. The manuscript is structured as follows. After a short preamble in Section 1, we unfold, in Section 2, four different perspectives on network model applications, spanning current network approaches to link cancer genomic dysregulation and cellular phenotypes using network mechanistic modelling and data-driven techniques, such as machine learning and deep learning (Section 2.1); the development of mechanistic and hybrid models to optimise cancer immunotherapies (Section 2.2); agent-based models of the physical tumour–host interactions (Section 2.3); and end-to-end systems that combine mechanistic models and machine learning for clinical decision support (Section 2.4). In Section 3, we discuss the bridging of the four perspectives and emphasise the need for and benefits that network theory and applications can bring to cancer research, both at the fundamental and translational level. We conclude with an outlook (Section 4) towards new research avenues that can stimulate the design and development of new theoretical and experimental approaches in the mathematical and computational oncology community.

## 2. A Multidimensional View on Networks in Cancer

### 2.1. Network Approaches in Cancer Genomic Research

Cancer is a genetic disease caused by the accumulation of somatic [2] and epigenetic [3] alterations throughout the lifespan of an organism. While most of these alterations are neutral and do not result in the loss of cellular fitness, a few mutations have a detrimental effect to the cell's normal functioning. These *driver* mutations increase an individual's susceptibility to developing cancer and, when combined with other driver mutations, might result in cancer onset and development. Despite some well-studied examples, e.g., hereditary breast and ovarian cancers driven by inherited mutations in the tumour suppressor genes *BRCA1* or *BRCA2*, most cancers are driven by the combined effect of multiple mutations that jointly dysregulate important cancer genes, such as oncogenes or tumour suppressors. Supporting this view, most cancers have been observed to present 2–8 driver mutations [4,5].

An additional fundamental observation has been the high level of heterogeneity in the observed somatic mutational landscape, where patients with the same type of cancer only share a small fraction of mutations. Indeed, the analysis of the genomic information from 50 cancer types catalogued by The International Cancer Genome Consortium (ICGC [6]) showed that only a small set of well-studied driver genes are frequently mutated, i.e., at frequencies higher than 10%, while the bulk of the mutational burden happens on infrequently mutated genes [7,8]. This variability can be understood when somatic mutations are analysed in the context of cancer networks. It then becomes apparent that mutations preferentially target pathways essential to controlling cellular homeostasis, such as pathways associated with signal transduction, cell cycle and apoptosis, DNA repair, etc.

The application of network approaches to model such genomic dysregulation has led to a better understanding of the tumourigenic mechanisms that underlie cancer onset and development [9]. In a cancer network model, nodes represent relevant components, e.g., genes, proteins, or cells, and edges account for interactions between these components. Ideally, a faithful characterisation of the network dynamics would require knowledge of the causal structure of the network. However, causal inference requires specific data, either time-series experiments or carefully planned perturbation experiments, both of which are not frequently available. In most cases, the available data are snapshots, such as the gene transcripts expressed at a particular time-point. An example of such a snapshot could be, for instance, the transcriptomic profile of a cancer patient at the time of diagnosis.

At the omics level, many techniques have been developed to infer undirected networks of gene or protein associations, also known as interactomes [10], using cohorts of patients' molecular profiles, typically transcriptomic or proteomic data. While each method has its own merits and weaknesses, it has been shown that no single method performs optimally across all data types and experimental conditions; however, *consensus*

*networks* built by integrating the predictions across different methods achieve excellent overall performance and robustness [11]. User-friendly applications have been developed to infer such consensus networks [12], as shown in Figure 1.

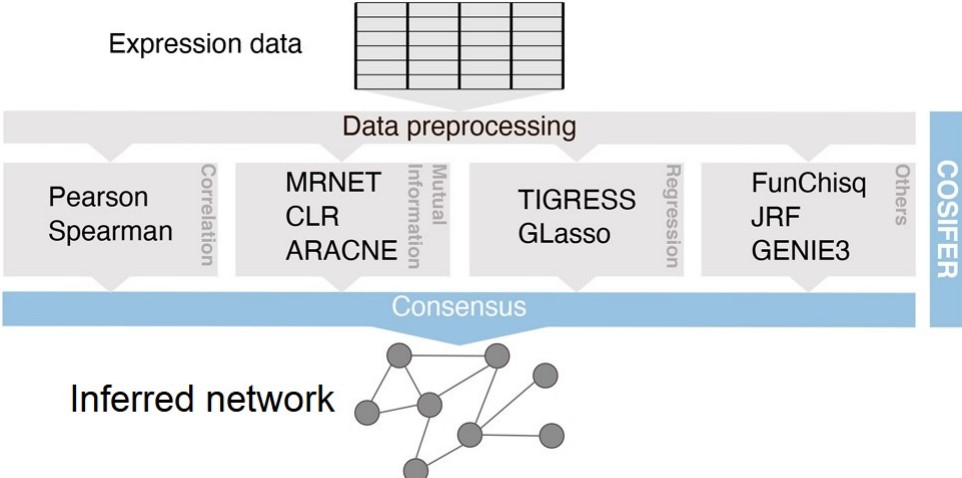

**Figure 1.** Network consensus inference. While many methods have been developed to infer undirected networks of molecular interactions using high-throughput data, the *consensus network*, i.e., the network built by integrating the predictions of diverse methods usually performs better and achieves higher robustness. User-friendly applications to build consensus networks are publicly available, such as, for instance, the consensus interaction network inference service—COSIFER [12], shown in the figure. COSIFER can be accessed as a Python package or through a web interface. A user can infer networks of molecular entities using a broad collection of different network inference techniques, or use state-of-the-art methodologies to generate a consensus network by integrating the predictions of each different method. Currently, the following network inference methods are implemented: Pearson and Spearmann correlation-based network inference methods [13]; ARACNE, a method based on mutual information [14]; the context likelihood of relatedness (CLR) algorithm [15]; MRNET, a maximum relevance/minimum redundancy network inference method [16]; regression methods such as GLasso [17], and the trustful inference of gene regulation using stability selection (TIGRESS) [18]; and a model-free method based on functional chi-squared and exact tests (FunChisq) [19], the nonparametric joint random forest (JRF) method [20], and the gene network inference with ensemble of trees (GENIE3) [21]. Figure reprinted from the work in [12] with permission from Oxford University Press.

Although in an interactome the causal dependencies between genes are not known, other important statistical and topological properties can be inferred and potentially correlated with important biological or clinical information. For instance, a human B cell interactome revealed a hierarchical, transcriptional control module, where MYB and FOXM1 acted as synergistic master regulators controlling the proliferation of B cells [22].

Additional network analysis can reveal important information by analysing node degree, betweenness centrality index, network density, the average number of neighbours, etc. [23]. Similarly, spectral graph analysis of multilayer networks can reveal similarities across cancer types [24]. To take advantage of the availability of prior knowledge regarding pathways, i.e., expert-based definitions of gene sets that share a common function, pathway-based approaches have been developed to statistically evaluate the possible over-representation or under-representation of mutated genes in those pathways [25,26]. More sophisticated approaches, which additionally take into consideration other factors, such as the magnitude of the genes' expression changes in the diseased compared to the normal state, the alteration type and position in the given pathways, or the type of interactions, have also been proposed [27].

To enhance the possibility of modelling genetic dynamics across scales, networks can be further coupled with mathematical or computational modelling techniques to

enable a detailed simulation of the network. For instance, ordinary differential equations (ODEs) [28,29] (Figure 2), stochastic models [30,31], or even hybrid models combining both continuum and stochastic descriptions [32,33] have been used to study the dynamical interplay of smaller subsets of important genes or cells, where the mechanistic interactions are defined according to prior knowledge.

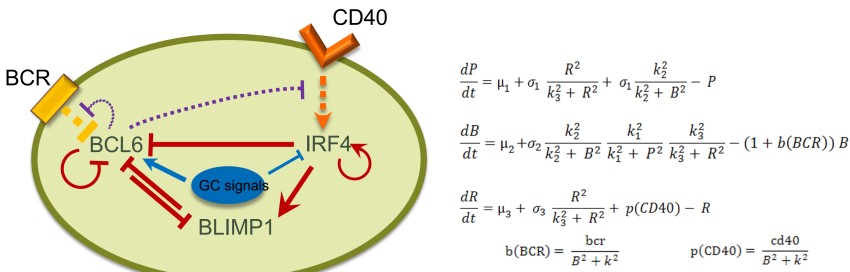

**Figure 2.** A system of ordinary differential equations derived from a prior knowledge network of the terminal differentiation of B cells enabled the investigation of underlying mechanisms of lymphomagenesis. (**Right**) The terminal differentiation of B cells is governed by 3 transcription factors, BCL6, IRF4, and BLIMP1, which mutually regulate each other in a highly coordinated and nonlinear fashion. In addition, two signalling pathways transmit regulatory information from the B cell receptor (BCR) and T cells (through the CD40 signalling cascade). (**Left**) Dynamical analysis of the ODE model associated with these molecular entities demonstrated that the coordinated interplay between both signals is necessary to promote the healthy development of plasma B cells [28]. Figure reprinted from the work in [28] with permission of the United States National Academy of Sciences.

The use of ODEs is sometimes limited by the need to infer kinetic parameters for which little information is typically available, as will be discussed in Section 2.4.2. To overcome this limitation, Boolean modelling approaches approximate genes as binary entities with only two possible states: *on* or *off*. This formalism results in discrete, deterministic, and parameter-free models that are easier to simulate and explain than ODE models, although the found dynamical properties might not always be consistent with the biological observations. Several extensions have been developed to render Boolean simulations more realistic, including iterative parameterisation, asynchronous updates, and the incorporation of continuous and stochastic elements [34–36]. In the context of cancer, Boolean modelling techniques have been applied to identify attractors corresponding to different cell phenotypes. Interestingly, mutations in key driver genes were shown to promote phenotypic transitions, which supported the discovery of routes of carcinogenesis [37]. The model was also used to evaluate the outcome of targeted cancer therapies. An interesting aspect of Boolean models is that they can be tailored to reproduce a particular biological sample such as a patient tumour, hence enabling the prediction of patients' clinical data [38], and the prediction of personalised drug responses [39].

More sophisticated machine learning and deep learning techniques have been designed that exploit cancer networks. For instance, network smoothing techniques that diffuse information over a network have been proposed to identify disease-causing genes [40], to identify cancer sub-types by integrating somatic tumour genomes [41], to integrate biological prior knowledge into machine learning workflows [42], or to gain insight into the molecular mechanisms underlying a sample's classification [43], just to name a few applications.

Beyond gene-level descriptions, machine learning models have been used to enhance network models with insights from tumour growth data [44], describe the phenotypic stages of a tumour from histopathological [45] or from radiomic features of contrast-enhanced spectral mammography images [46], and even suggest personalised therapy sequences tailored to each patient [47], as further discussed in Section 2.4.2. More sophisticated approaches for data integration have been achieved by constructing networks of

samples, i.e., networks of patients, for each available data type and then efficiently fusing these into one network that represents the full spectrum of underlying data [48].

*2.2. Network Approaches for Cancer Immunotherapy Optimisation*

Cancer onset and development are strongly influenced by the immune system, which might either promote or attenuate tumourigenesis and strongly influences therapeutic outcomes. While during the early phases of tumour development the immune system can identify and eliminate abnormal cells, chronic inflammation is associated with tumour development, progression, metastatic dissemination, and treatment resistance. Furthermore, chronic inflammation can be immunosuppressive and extinguish antitumour immune responses that emerge as a response to the expression of neoantigens, i.e., tumour-specific antigens expressed by cancer cells as a result of newly acquired mutations. The antagonism between inflammation and immunity can strongly affect the outcome of cancer treatment, and its implications are only now starting to be investigated [49].

In recent years, cancer immunotherapies focused on reactivating the immune system are emerging as powerful therapeutic options. Immunotherapies generally seek to redirect the immune system's natural cytotoxic activity against malignant tissues, either by reactivating exhausted T cells or by blocking immune checkpoint regulators that prevent immune cells activation. One of the most promising approaches are chimeric antigen receptor (CAR) T cell therapies, where patients' T cells are engineered to express a surface receptor designed to bind a cancer neoantigen, expanded *ex vivo* and re-infused into the patient [50]. CAR T cell therapies have resulted in durable remissions for some patients with certain types of cancers [51], although much work remains to be done to understand and overcome current challenges, such as gaining a better understanding of why some patients do not respond to the therapy, investigating and reducing severe associated toxicities, improving infiltration in solid tumours, extending the approach to a larger set of cancer types, etc. [52].

Many of these questions can be investigated with the help of network models. For instance, to accurately simulate the activation and therapeutic effect of CAR T cells, models need to jointly account for the binding of a cancer neoantigen by a CAR, and the subsequent activation of the T cell intracellular signalling pathways leading to the T cell antitumour function. Supporting these efforts, several network-based models have been developed to capture T cell signalling dynamics [53]. Some of them integrate different modelling approaches, such as logical, constraint-based, agent-based, and ODE models, to capture processes taking place at different spatial scales [54].

It is important, however, to keep in mind two considerations: First, although logical models and constraint-based models can provide good approximations of the qualitative behaviour of a biochemical system without the burden of parameter optimisation, obtaining a detailed description of a network's dynamics, especially in the presence of nonlinear effects associated with irreversible transitions, bifurcations, or attractors, usually requires the use of a modelling formalism based on continuous differential equations [55,56]. Second, the activation mechanism of engineered T cells may differ substantially from T cells carrying native T cell receptors, and therefore models specifically designed to understand these differences are crucially needed.

In addition, the accurate modelling of complex immune responses is likely to require the development of multi-cellular mechanistic models that can capture the dynamical interplay between different types of immune cells. For instance, models to explain B cell development and its dependence on effective interactions with T cells and follicular dendritic cells have already been developed based on ODEs [28], stochastic hybrid models [33], and agent-based models [57], although their integration in the context of larger immune multi-cellular frameworks remains to be realised.

As the field moves forward and new and more informative datasets are being generated, the field of mathematical and computational oncology might also have to move to the development of hybrid models that combine traditional mathematical approaches, such

as ODEs, partial differential equations and stochastic models, with machine learning and deep learning frameworks. In that sense, deep neural networks (DNNs) can be viewed as a coarse-grained iterative process, where the first layers learn low-level features that are combined in a nonlinear fashion into more abstract and complex features. The final prediction is often made by using simple classifiers that exploit the high-level features, thus combining all the information extracted from the sample in a highly nonlinear fashion. Indeed, DNNs have been shown to be powerful feature extractors that can effectively mimic the re-normalisation process [58], an approach that has been used in physics to accurately describe the macroscopic behaviour of a system without knowing the exact microscopic state of all its components. Similarly, by summarising low-level features into increasingly complex higher-level features that are predictive of macroscopic properties, DNNs can be seen as re-normalisation machines. Based on this idea, hybrid models can be developed to mechanistically integrate abstract DNN-generated features with observable variables that quantify the molecular and cellular responses after, for instance, T cell activation.

The advantage of hybrid models is that different modelling approaches can be used to model each system's components based on its symmetries. For instance, DNNs are especially adept at modelling raw low-level data, and can therefore be used to model systems that are too complex or lack the necessary symmetries to be described with mathematical or statistical approaches, such as aggregates of cells or interactions between protein complexes. For instance, a multimodal deep learning model was used to successfully predict the binding affinity between a T cell receptor and an epitope [59], a system that would have been very difficult to model through mechanistic approaches. Other components, however, can be more efficiently described using mechanistic formalisms depending on informative biological observables that can be directly quantified. An example of the latter are, for instance, protein signalling cascades or gene regulatory networks. In both cases, the activity and expression level of the involved actors can be directly quantified, and a wealth of prior knowledge about molecular and cellular interactions can be used to further constrain the topology of the interaction networks. Communication between different compartments can be designed in a modular fashion, with communication channels that summarise the collective state of a compartment and send the information to the neighbouring compartment. Alternatively, new hybrid networks can be designed that merge both measured and DNN-derived variables and seamlessly simulate the ensemble together.

### 2.3. Network Approaches in Mechanistic Modelling

Computational modelling/simulation constitutes a fundamental tool in the scientific method. It complements the other existing approaches, namely, the mathematical/analytical approach and the experimental approach. The advent of high-performance computing resources has enabled a dramatic acceleration of computer simulations and revolutionised many fields, including biological, e.g., neuroscience, systems biology, oncology, etc., as well as non-biological fields, e.g., financial economics, mechanical engineering, etc. The computational approach offers a number of advantages, particularly in combination with the experimental approach, as it can reduce the number of experiments needed, serve as a test bed for competing hypotheses, and yield experimentally testable predictions. As such, it is an attractive tool for cancer research, particularly in modern times where large and complex datasets are collected.

In principle, computational models can be divided into *mechanistic* and *non-mechanistic* models. The former type is characterised by the presence of assumptions on the governing causal relationships, while the latter remains agnostic on these, but aims to describe and predict observable features. Both types have evident advantages and disadvantages, and should be chosen carefully depending on the available information and purpose of the study. It has been shown that computational models based on mechanistic as well as non-mechanistic interaction networks can provide valuable insights into dynamics of cancer. In many cases, the lack of (a priori) knowledge renders it difficult to initiate the modelling with assumptions on these dynamics. In such cases, it is appropriate to treat the system as

a black or grey box. Machine learning-based models that are trained using large amounts of experimental data are attractive candidates for this approach [60], particularly due to the availability of high-performance computers that minimise the required simulation time.

Here, we describe examples of mechanistic and non-mechanistic models in computational oncology. Both types of computational models comprise network-based formalisation, including networks of interactions. Notably, mechanistic models can be differentiated into two types, namely, continuum models and discrete models [61].

### 2.3.1. Continuum Modelling

Continuum models incorporate mathematical methods based on differential equation models, namely, ordinary differential equations (ODEs) and partial differential equations (PDEs), allowing to quantitatively capture aspects of cancer dynamics. In contrast to ODE-based continuum models, PDE models enable the modelling of cancer dynamics in space. Given the well-established importance of spatial aspects in the outcome of cancer treatments [62,63], the inclusion of these is a crucial feature for computational models of cancer progression. One of the earliest examples where spatial dependencies were considered is the study of modelling interactions at the tumour–host interface [64]. Since then, numerous studies have yielded important insights into cancer progression and the impact of therapy. However, these models have the shortcoming that they do not allow to incorporate genetic dynamics and behaviours of individual cells. However, these issues can be addressed with so-called discrete models. Given the focus of our current study on network-based and multi-scale models, we will elaborate on discrete models in the following sections.

### 2.3.2. Discrete Modelling

With more detailed, high-throughput data it is becoming viable to make concrete assumptions on mechanistic interactions, across all spatial levels. Combined with the presence of dedicated software, e.g., PhysiCell [65] or BioDynaMo [66], these data enable the creation of computational models that have high explanatory power. Along those lines, agent-based models (ABMs) are arguably the most powerful type of discrete models for tissue dynamics [67]. Indeed, ABMs are becoming more and more prevalent in cancer research [68], as their particularly high computational requirements have become less problematic as they used to be [69]. These models allow to specify in a "bottom-up" approach not only intracellular, metabolic, and genetic dynamics, but also mechanical intercellular interactions in 3D space [70]. Therefore, they constitute a well-suited framework for integrating network-based dynamics. It is therefore not surprising that ABMs are used not only for cancer research, but also in many other disciplines such as developmental biology [71], synthetic biology [72], cell biology [73], or public health [74].

Generally, networks in ABMs can be formulated on three levels: the sub-agent level, the agent-level, and the extra-agent level (Figure 3). In the context of biological simulations, the sub-agent level could for instance denote the sub-cellular level of molecular interactions, gene regulatory networks or metabolic networks. Their dynamics can affect other cells but only indirectly by exerting impact via the host cell within which they reside.

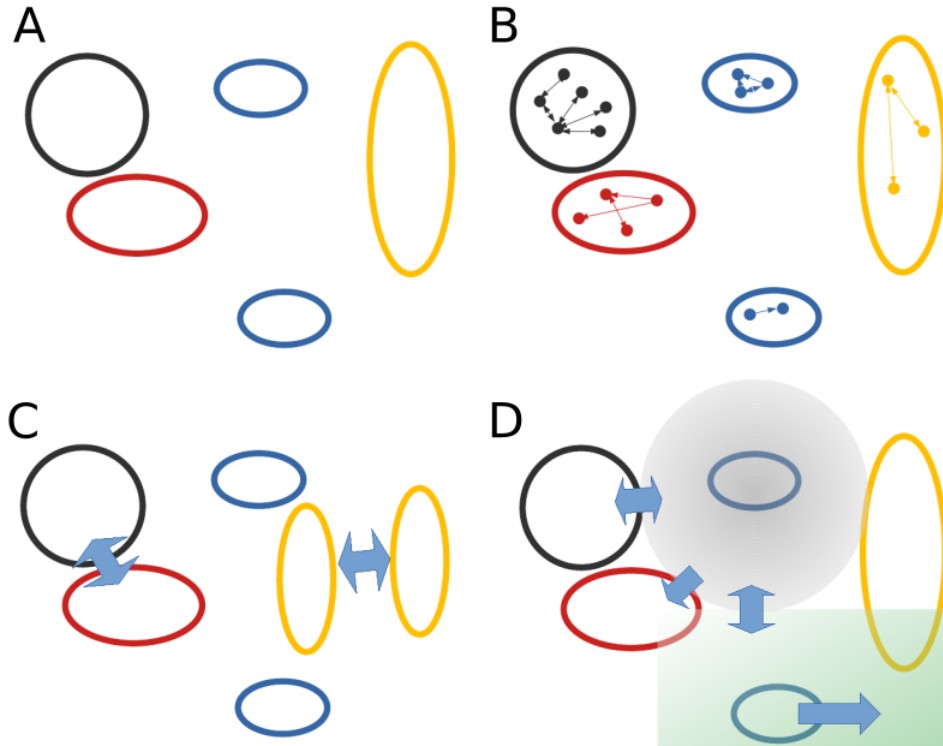

**Figure 3.** Levels of agent-based cancer modelling. Agent-based computational models of cancers can be formulated on three different levels. (**A**) Agent-based cancer models entail elements that act according to genetically defined rules that define their behaviour. These elements reside within physical space, which can be 2D or 3D. (**B**) Networks within agents can be modelled, accounting for gene regulatory networks, metabolic pathways, or other dynamics. (**C**) Agents themselves participate in interactions that can also be formulated as networks. For instance, cell bodies interact with one another, or they proliferate and give rise to new elements that continue interactions. (**D**) Finally, network models can also be formulated in the extra-agent space. Along those lines, extracellular substances can interact and give rise to reaction–diffusion systems [75]. Diffusible substances and other extracellular elements, e.g., surgery tools, radiation, temperature, etc., also constitute part of the extra-agent network level.

On the intra-agent level (Figure 3B), networks of interactions on the sub-cellular scale are modelled. This includes gene regulatory networks, metabolic networks, and protein–protein interactions (PPIs). On the agent level (Figure 3C), the networks of interactions usually reside within the extracellular neighbourhood. These can comprise mechanical interactions, for instance, when cells adhere to or push one another. Alternatively, these can be behavioural interactions, where cells detect nearby cells and interact via receptors and ligands. The crucial dynamics between cancer cells and immune cells described earlier can be attributed to this level. Finally, on the extra-agent level (Figure 3D), processes occur without requiring the active action of cellular components (agents). This can be for instance the reaction and diffusion of interacting substances in extracellular space, as well as oxygenation across space due to the effects of vasculature [76]. As drugs or radiation impact cancer cells via the extra-agent level, ABMs of therapeutic interventions, e.g., chemotherapy or radiotherapy, need to take into account the transmission on this particular level, as well as its impacts on the intra-agent level [77,78].

Notably, networks of interactions on these three levels differ not only in terms of their spatial scales, but also with regard to their temporal scale. Along those lines, certain physical, molecular interactions such as those in PPIs act on shorter timescales than, for instance, functional interactions governing gene regulation. The agent-based approach offers a powerful framework to integrate such heterogeneous spatial and temporal networks into coherent multi-scale models. To this end, ABMs require networks of interactions to be

directed (uni- or bidirectional), because they are mechanistic and so need to have defined directions of action. A schematic visualisation of a cancer tissue ABM is shown in Figure 4.

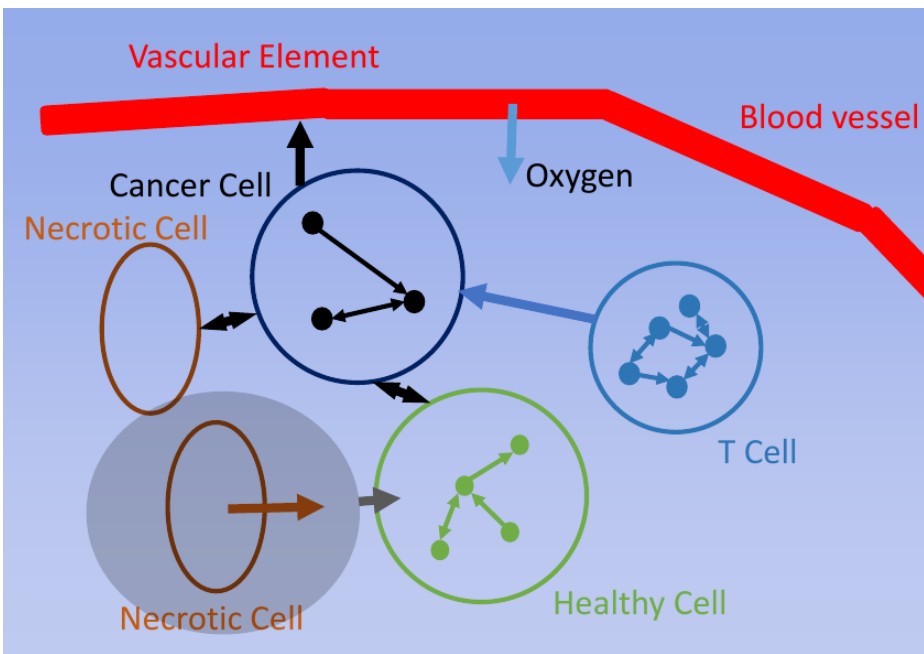

**Figure 4.** Example agent-based model (ABM), constituting multi-scale networks of interactions in cancer growth. Different types of cells and elementary "agents" constitute the modelled system. In this case, the cells can be healthy (green), cancerous (black), necrotic (brown), T cells (blue), and vascular elements (red). These behave as autonomous agents that interact with one another: necrotic cells release intracellular components into the extracellular environment (black, circular and transparent area). Therefore, they interact with different cells. Analogously, cancer cells can interact with healthy cells in the tumour microenvironment, or promote vascularisation (black arrows). T cells can kill cancer cells (dark blue arrow), and oxygen provided by the vasculature (light blue arrow) has important effects on healthy as well as cancerous tissue (blue, shaded background area). Additionally, intracellular regulatory networks govern biological behaviours.

ABMs have been successfully employed to elucidate various processes in cancer growth and interactions of cancer cells with immune cells. For instance, the authors of [79] employ agent-based modelling to model cancer growth and interactions between cancer cells and immune cells. Their model reproduces certain realistic spatial patterns, and could potentially be used to predict immune checkpoint blockade treatment outcome. Notably, their model reproduces the observed correlation between mutational burden and the response to immune checkpoint therapy [80] (Figure 5). In particular, it yields quantitative estimates on suitable antigen strengths that have a high impact on the number of cancer cells. Along those lines, we anticipate a stronger utilisation of agent-based modelling for pharmaceutical research in the future: it constitutes a cost-effective method for predicting the impact of different spatial, temporal, and dosage parameters in oncology.

One of the main challenges in the agent-based modelling of cancer is combining datasets to create an integrated and coherent picture. This is a complex task because individual experimental labs and research projects usually focus on highly specific questions. Overall, deeper and more far-reaching questions can be addressed with the availability of datasets spanning across different scales, and comprising information on cells, the immune system, the extracellular matrix, and other important factors.

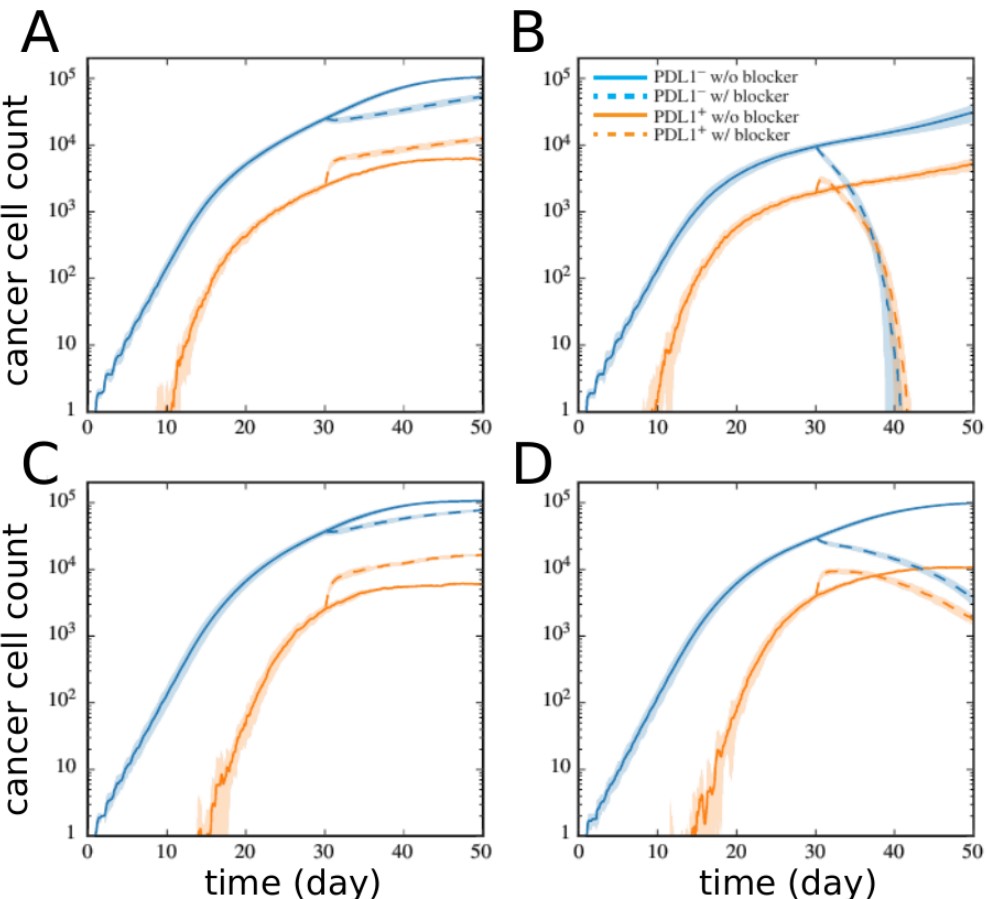

**Figure 5.** An ABM reproduces the correlation between mutational burden and response to immune checkpoint therapy. In the study of [79], the number of PDL1 positive (PDL1$^+$) and negative (PDL1$^-$) cancer cells is modelled. The impact of immune checkpoint therapy was simulated via the reduction of cytotoxic T cell suppression. For cancers with low mutational burden (**A**,**C**), immune checkpoint therapy has little impact, or can even increase the cancer cell count. In contrast, the computational model predicts a positive treatment effect in cancers with a high mutational burden (**B**,**D**). Dashed lines indicate treatment scenarios starting at day 30. Solid lines indicate cell numbers in untreated patients. Figure adapted from [79].

## 2.4. Learning Mechanistic Interaction Networks

The following dimension of networks in cancer examines the various physical interactions, including cancer kinetics, tumour-immune systems interactions, and treatment outcomes, through the lens of processes and dynamics at the system and clinical application levels [81]. Such processes are described by coupled interactions from a network perspective, combining the detailed modelling described in Sections 2.1 and 2.2 and the computational framework described in Section 2.3 towards clinical use. This section unifies the insights and outcomes of using network analysis to study both bottom-up and top-bottom interactions in cancer. It further elaborates on a possible framework and research strategies in this area by emphasising and discussing the multiple facets of such a perspective across scales.

Fundamentally, the dynamics governing cancer development are informed by quantitative measurements that describe tumour growth, host cell encounters, and drug transport in order to predict a patient's outcome [82]. For instance, spatio-temporal modelling of intracellular pathways associated with cancer can now, with the power of machine learning, be linked across scales to tissue level manifestations of malignant neoplastic processes [83]. At the same time, this approach can strengthen the links between cancer biology and its

physics through data-driven learning systems, which hold the potential for the discovery of new drugs and treatment strategies [84].

This perspective advocates the combination of mechanistic modelling and machine learning that goes beyond measurement-informed bio-physical models and towards personalised disease evolution profiles learnt from clinical oncology data [85]. Tackling both bottom-up and top-down interactions, such a framework could offer a better understanding of the causes and consequences of (physical) interactions in cancer and their connections to the biological hallmarks of cancer broadly described in Sections 2.1–2.3.

### 2.4.1. Grounding the Learning Mechanistic Interaction Networks

Physical interactions of cancer cells with their environment, e.g., local tissue, immune cells, or drugs, determine the physical characteristics of tumours through distinct and interconnected mechanisms. For instance, cellular proliferation and its inherent abnormal growth patterns lead to increased solid stress [86]. Subsequently, cell contraction and cellular matrix deposition modify the architecture of the surrounding tissue, which can additionally react to drugs [87], modulating the stiffness [88], and interstitial fluid pressure [89]. However, such physical characteristics also interact among each other, initiating complex dynamics as shown in Section 2.3. Learning mechanistic interaction networks can capture such complex dynamics and can unfold a network-based paradigm for modelling, computation, and prediction. They can extract the interactions among the various entities, e.g., tumour, host cells, and cytostatic drugs, by learning the physics of their interactions for producing informed outcome predictions. A simple, biologically grounded, example is depicted in Figure 6.

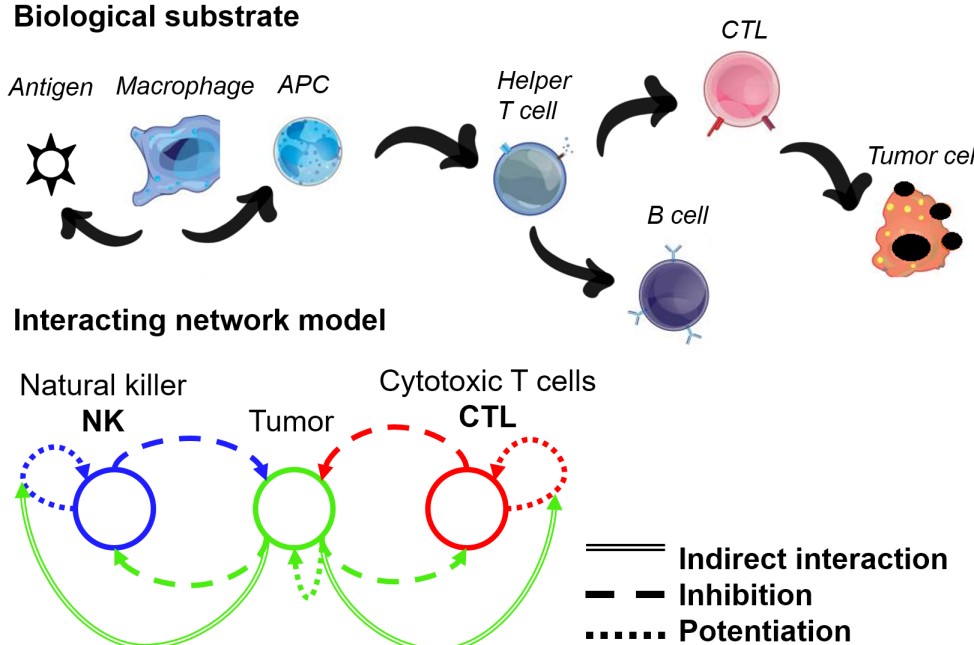

**Figure 6.** Biological grounding of learning mechanistic interaction networks. The activation of the adaptive immune response in the presence of the tumour cell is an example of a biological substrate. The macrophage (a type of Antigen-Presenting Cell (APC)) recognises the non-self (tumour carried) antigen and engulfs it. Subsequently, the APC presents the phagocytised cell to immature T cells which begin to proliferate, triggering other immune cells (i.e., B cells and natural killer cells (NKs)) and the killer T cells (CTLs) responsible for recognising and killing malignant cells. Such interactions are captured in the interacting network model where each of the entities (i.e., NKs, CTLs, and tumour) is described by a circle interconnected through edges. Each of the circles describes the entity dynamics (e.g., tumour proliferation) and the edges the type of interactions (e.g., inhibition, potentiation, and indirection interaction, respectively).

This simple interaction network can then describe and predict specific phenomena and interactions among the tumour and the various arms of the immune system broadly described in Section 2.2. For example, starting from the interaction network in Figure 6, the underlying computation and learning functionality could, for instance, simultaneously capture the power-law tumour growth under immune escape [90] and the potentiation—inhibition model of Natural Killer Cells (NKs)–tumour interactions [91], while exhibiting the known overlapping Cytotoxic T Lymphocytes (CTLc)–Natural Killer Cells (NKc) regulation [92]. Such a basic instantiation is depicted in Figure 7.

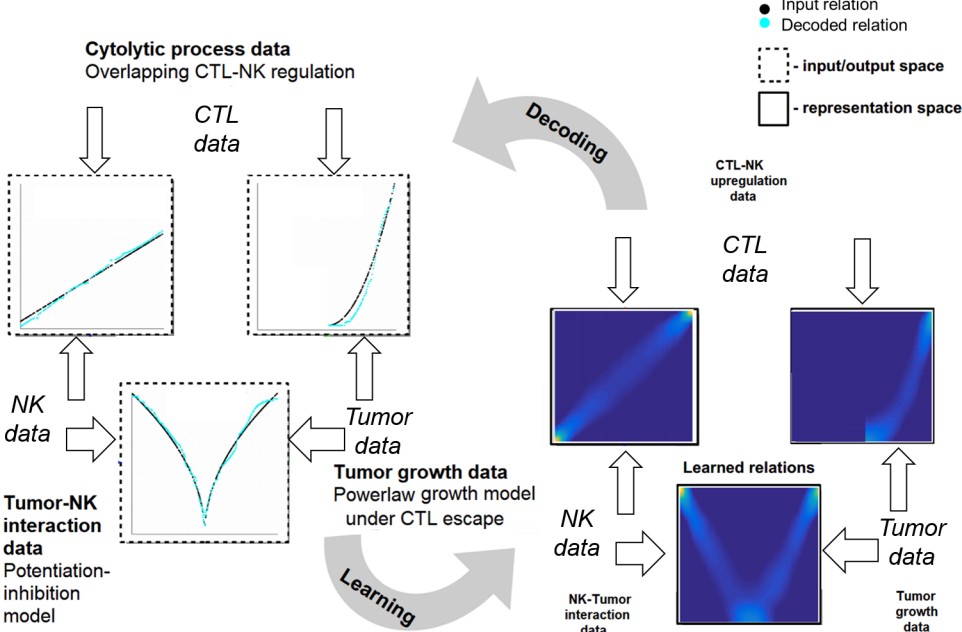

**Figure 7.** A generic computational substrate of learning mechanistic interaction networks. For a sample context, we consider the interactions between the two arms of the immune system (i.e., natural killer cells (NKs) and killer T cells (CTLs)) and the tumour. The input relation (marked in black in the left panel) describes the pairwise interactions among the three entities in the input space. These temporal patterns of interactions among the three entities are the input to the learning system. After learning the correlations among each pair of entities, the underlying (learnt) relations are decoded from the representation space of the network. The system is able to simultaneously learn the CTL-NK regulation under the constraints induced by the potentiation–inhibition model of the innate immune system and the power law growth model of the tumour under CTL escape. The system output (marked in turquoise) is compared with the original data. Note that the system had no prior information about the data statistics and learnt the interaction patterns without supervision.

As shown in Figure 7, the learning mechanistic interaction networks offer the means to learn the physical laws/relations governing the tumour–immune interactions from clinical data (see Figure 7 Decoded vs. Input relation) in order to make predictions on the effects of modifying the nonlinear pattern of interactions among the tumour and the immune system (after learning). Such networks are generic, data-driven, and not constrained to a certain cancer type or cell line. Their networked structure allows to easily interconnect multiple computational maps for more biological components [93] or a different granularity or scale [94] of representation of the underlying interaction physics.

### 2.4.2. Instantiations of Learning Mechanistic Interaction Networks

In this section, we go through different studies that support the unified view of learning mechanistic interaction networks. The purpose of this section is to demonstrate that there is a large body of research describing learning mechanistic interaction networks, from processes and dynamics [45] at the system level to the clinical application level [47].

More precisely, we will focus on four problems where learning mechanistic interaction networks are applied, namely, extracting cancer kinetics, phenotypic staging, therapy sequencing, and therapy outcome prediction.

**Extraction of cancer kinetics.** Neoplastic processes are described by complex and heterogeneous dynamics. The mechanics of the interaction of neoplastic cells with their environment describes tumour kinetics and is critical for the initiation of cancer invasion [95]. In this context, learning mechanistic interaction networks provide data-driven models [44,96] capable of unsupervised learning of cancer growth curves within, i.e., breast cancer cell lines MDA-MB-231 [97] and MDA-MB-435 [98], and between cancer types, i.e., lung [99], breast, and leukaemia [100]. This is achieved through computational mechanisms that learn the temporal evolution of the tumour growth data obtained either from imaging, e.g., in [97,101]; caliper [99]; or microscopy [99,102], along with the underlying distribution of the input space.

**Tumour growth and phenotypic staging.** Ductal Carcinoma In Situ (DCIS), a significant precursor to invasive breast cancer, is typically assessed using mammogram diagnosis [103]. But this approach is not accurate, i.e., initial cancer cells are typically classified as microcalcifications. This is mostly due to the poor understanding of the DCIS growth process [104,105], its highly genomic/proteomic- and microenvironment-dependent phenotype [106,107], and its cell volume modifications during proliferation and necrosis [108]. In this context, learning mechanistic interaction networks have already shown the potential they hold for extracting breast tumour growth patterns by learning the underlying mathematical relations/functional dependencies of phenotypical transitions of cancer cells in order to predict the percentage of the breast tissue affected by the tumour (as for instance in DCIS) [109]. Such approaches used timeseries of raw immunohistochemistry and morphometric data to infer the mathematical relations [45,110] describing the DCIS phenotypical transitions from quiescence to proliferation and from proliferation to apoptosis, consistent with experimental data [102]. This way the model could precisely estimate the tumour size and thus advise the surgeon how much tissue should be surgically removed [111,112].

**Chemotherapy-surgery sequencing.** Neoadjuvant treatments promote improved resectability by decreasing the initial tumour size and increasing healthy tissue-conserving surgery rates [113]. This evaluation enables the therapist to cease inefficient treatments or transition to a different regimen for better results [114]. Neoadjuvant chemotherapy is typically the prevalent procedure [115]. However, patients without a pathological complete response (PCR) after the neoadjuvant intervention will, typically, follow the adjuvant or therapy combinations schemes [116]. Overall, it is reported that the long-term outcome of both therapy strategies (i.e., neoadjuvant and adjuvant) strategies is nowadays similar. The most relevant question at the moment is to ask what is the best strategy for a particular patient? This question describes the combined interactions of multiple patient-specific factors, such as the tumour growth curve, the parameters that determine the chemotherapy response, and the drug pharmacokinetics, that impact, and typically tailor, the course of therapy (i.e., chemotherapy and/or surgery) [117]. There is a series of very promising studies [47,90,110] employing a data-driven approach in which individual patient data describing tumour growth, e.g. histology, imaging, and chemotoxic drug effects, i.e., pharmacokinetics and drug interactions, are used in combination to extract the optimal sequence of therapy. Such systems can explicitly manage the biological variability of tumours, the limited amount and types of patient data, and the variability in the chemotherapeutic drug response, in a data-driven way that captures the interaction between tumour and chemotherapy [118].

**Therapy outcome prediction.** The differences between research conducted in the laboratory on monolayers of cancer cells and treatments undertaken in real patients or in animals are a major challenge confronted by oncologists when designing chemotherapy strategies [119]. Various chemotherapeutic drugs, when administered to regulated monolayers, they produce excellent outcomes, but they underperform in animal models [120]. Furthermore, *in vivo* cancers have built strong and adaptive resistance mechanisms to

systemic chemotherapy [118]. A first reason for this is a very interesting phenomenon, namely the fact that as tumours develop their growth factor decreases. Therefore, given that the chemotherapy regimens are designed to select cells that are quickly dividing, treatment resistance develops inherently [121]. Second, random genetic alterations combined with natural selection can result in tumours that are resistant to almost any chemotherapy [122]. A tumour's microenvironment, e.g., acidosis, increased interstitial fluid pressure, hypoxia, tissue density, drug wash-out, or obstructed blood flow, is a key player modulating drug's infiltration and killing potential [123]. Learning mechanistic interaction networks are also employed in therapy outcome prediction [96,124] and survival analysis [125]. Such approaches use machine learning algorithms to extract tumour growth patterns in both therapy-free and when following neoadjuvant chemotherapy regimens. By mining one-dimensional tumour growth time series data (i.e., tumour volume measurements acquired in different ways: MRI, caliper, and ultrasound), such systems learn the tumour growth dynamics and the tumour's response to chemotherapy. Such approaches can predict chemotherapy effect more accurately than traditional mechanistic tumour growth models [126]. These contributions strengthen and open new research avenues of exploration towards the design of data-driven therapies that capture and exploit differences observed among the patients for optimised interventions [127]. Finally, nomograms are known tools to help patients and physicians make important treatment decisions. Based on information collected from thousands of patients, this tool can be used to predict cancer-specific survival [128] or assess risk based on patient-specific characteristics and tumour evolution [129]. Although they provide excellent graphical depictions, nomograms are limited in terms of capturing a large number of variables and their interactions in predictive models [130]. Dynamic nomograms are a novel translational tool to help address this issue [131] especially for cancer evolution models containing modifiable risk factors, where the combination of nomograms and learning networks provide a promising future approach to follow [132].

As we have seen up to now, learning mechanistic interaction networks allow for *in silico* modelling and analysis of cancer growth and therapies. Furthermore, they can (1) describe their effects on the physics of the tumour–host–drug interactions [84]; (2) understand the development and evolution of tumour–host–drug interactions over time [133]; and (3) extract the interaction patterns under possible perturbations [134], such as drug re-dosing, while characterising disease dynamics across scales [135]. Such an approach nicely complements the low-level genetic processes modelling and therapy design approaches described in Sections 2.1 and 2.2 and the physics modelling approaches from Section 2.3 with a set of available and powerful tools offered by machine learning.

## 3. Unifying the Dimensions

Conventional cancer treatments are often sub-optimal due to a limited understanding of the underlying molecular and cellular mechanisms driving the disease and insufficient accuracy when predicting how an individual patient is going to respond to an administered therapy. Yet, in the last decade advances in both fundamental and clinical oncology and data science have clearly illustrated the need for a computational oncology-based approach to investigating and treating cancer. This new approach sees cancer as a combination of mathematical, physical and biological problems, rather than a strictly biological one.

Tumour evolution and response to treatment depend on a plethora of factors, including physical constraints, nutrient dynamics, the tumour–host interaction, vasculature development, drug delivery processes, etc. These processes are in turn influenced by the genomic, phenotypic, and microenvironment changes fuelled by cancer cell proliferation, and ultimately determine the tumour's response to treatment. Mathematical and computational models that describe the genetics of tumour initiation, the physics of a tumour's growth patterns, and a tumour's response to therapies, among others, are becoming a useful instrument for physicians in the oncology practice. Capturing the multi-factorial interactions among tumour and host, network theory models hold the promise of providing

an unified framework to model, analyse, and predict a tumour's evolution and response to therapy.

Each tumour is unique and should be treated as such. Although the pattern of interactions among tumour and host is similar, the peculiarities of the interactions at each level, i.e., cell, tissue, and whole body, are patient-specific. Network theory enables scientists and physicians alike to account for multiple scales of tumour evolution under treatment, from the cellular to the tissue scale, and ultimately, to predict the effect on the whole body. Our review integrates this view into a unified perspective that combines detailed molecular descriptions, as discussed in Section 2.1, cellular descriptions, such as agent-based modelling described in Section 2.3, and hybrid mechanistic and machine learning approaches, discussed in Sections 2.2 and 2.4, respectively.

As we have seen in the four sections, network theory can provide tools to:

- enable the investigation of fundamental mathematical, physical and biological principles derived from experimental data;
- fuse data from different sources, such as genetics, imaging, pathology, and mammography, to capture patterns at multiple scales to characterise tumour evolution;
- predict a tumour's evolution after a specific treatment in a personalised manner, such as immunotherapy or conventional drug administration; and
- alleviate over-treatment, where a patient receives treatments or invasive procedures that might not be necessary.

Our unified perspective introduces a clinically relevant approach that exploits quantitative measurements, including molecular, phenotypic, and diagnostic information, to generate robust predictions that are personalised and go beyond current subjective and heuristic assessments. We advocate for the combined use of mathematical modelling, network theory, and machine learning to optimise therapy design in clinical oncology. We suggest that, given the demonstrated potential such an approach provides, the next focus should be on adapting the models for clinical use. The framework we propose considers the development of practical mathematical and computational tools that can be used in the clinic to predict treatment outcome for each individual patient prior to administering a treatment.

This section is an attempt to unify the discussed perspectives and dimensions of networks in cancer. Across temporal and spatial scales, network theory models and tools capture the interactions among the multiple entities involved in cancer initiation, development, and therapy, as depicted in Figure 8.

Going beyond the cross-layer combination of methods and analysis in Figure 8, we can provide a more structured view on how the different methods and network systems contribute between and within the layers towards reaching the clinical goal. From the complexity point of view, ODEs can capture known gene dynamics and interactions at the cancer genetics level but with the risk of not capturing the interpatient variability. On the other side, consensus networks mine large datasets that support generalisation by capturing long-range mutations, but are complex and need very elaborate data processing pipelines. ABM networks offer typically a trade-off when it comes to complexity, exploiting both the biophysical knowledge and the available data to describe tumour dynamics. From the explainability point of view, ODE methods and ABM methods dominate use cases at both cancer genetics and cancer physics levels. Yet, their reductionist tendencies incline to offer machine learning methods the pole-position in cases where large heterogeneous data sources are available. Additionally, today's advances in explainable and trustworthy machine learning will minimise this gap allowing such methods to be more openly adopted in the clinic. Independent of data quantity, complexity, and explainability features of the individual network-based methods, the current technological and data availability landscape motivates more research to be conducted towards hybrid systems, where physics-informed machine learning is a key tool in the design of clinical decision support systems. We anticipate that the unification and combination of mechanistic modelling and machine learning across scales will be of major significance for clinical decision-making.

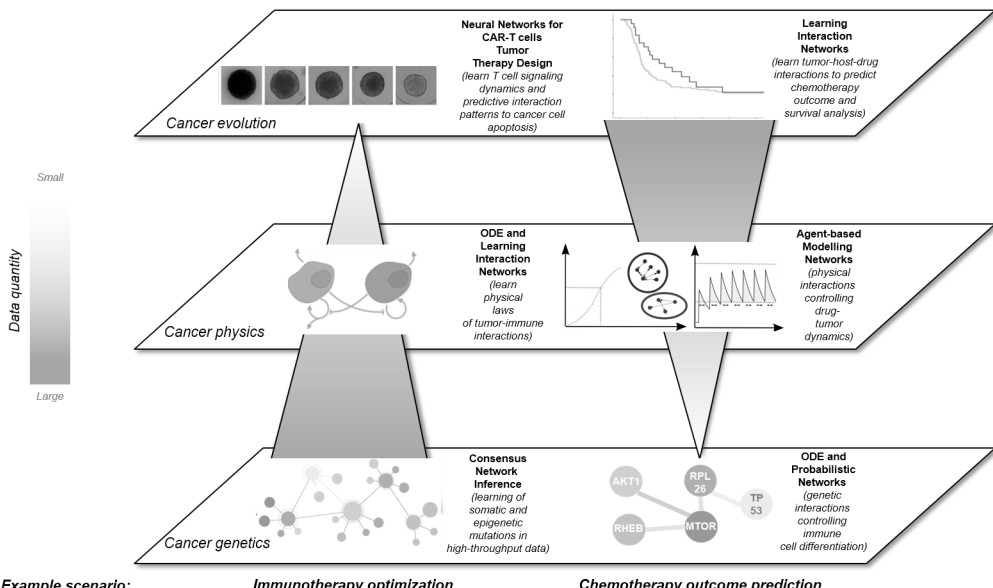

**Figure 8.** Unifying Perspective for two relevant clinical tasks. **Immunotherapy optimisation**: Presented with large amounts of high-throughput data, at the cancer genetics level, consensus networks and gene regulatory networks (described in Section 2.1) enable a twofold investigation of the genomic dysregulatory events underlying cancer. At the next level, the one of cancer physics, ODE modelling, and learning interaction networks (described in Sections 2.3 and 2.4) extract tumour-immune interactions patterns to quantify the impact of CAR T cells dynamics. Finally, close to the clinic, the cancer evolution level is described by the use of powerful deep learning systems capable of extracting T cell signalling patterns for clinically relevant tumour evolution to apoptosis (described in Section 2.2). **Chemotherapy outcome prediction**: A typical therapy approach in most cancer types, chemotherapy considers well-defined scheme designs. Typically defined by small amounts of cancer genetic data, chemotherapy designs model genetic interactions, such as the BRCA1 and its network of interacting partner genes, through ODE approaches and probabilistic networks (described in Section 2.2). Such tools can enable, additionally, the identification of relevant interactions controlling immune cell differentiation. With histopathology information available, at the cancer physics level, agent-based network simulations (described in Section 2.3) can model the physical spatio-temporal interactions patterns among tumour and drug and can generate insights into processes associated with cancer invasion, pharmacodynamics, and therapy planning. Finally, capable of capturing the complex tumour–host–drug interactions, learning mechanistic interactions networks, exploit all clinical data available and predict, simultaneously, tumour evolution, drug pharmacokinetics, and chemotherapy outcome, as discussed in Section 2.4.

## 4. Outlook

Despite the great progress in both clinical oncology and data science, more research is needed to bring mathematical and machine learning models to patients in the clinic. The research and clinical oncology community is facing pressing scientific and practical problems that need to be overcome. For instance, we need a better understanding beyond mechanistic models of the emerging biological and physical phenomena that govern tumour-host interactions in the context of new therapeutic strategies, such as targeted drugs, immunotherapy, metabolic therapy, and nanomedicine. This is only possible when data-driven approaches, such as machine learning and deep learning, are combined with classical mechanistic approaches to leverage both large and small heterogeneous datasets to produce personalised patient recommendations.

Future multi-scale and multicellular models trained on new multi-omics high-throughput data will undoubtedly better recapitulate the molecular and cellular dynamics that drive cancer onset, progression, and response to therapy, as we have seen in Section 2.1. The combination of different modelling approaches, described in Sections 2.3 and 2.4, in-

cluding both mathematical and data-driven techniques, will be essential, as it is unlikely that mechanistic models, whether based on continuous, discrete or qualitative frameworks, will be expressive enough to accurately and faithfully reproduce the complex landscape of cancer phenotypes. For instance, as discussed in Section 2.2, the activation of cytotoxic T cells relies on carefully tuned molecular recognition events between T cell receptors and cancer epitopes, which are unlikely to be modelled precisely enough through mechanistic approaches.

The path to move forward might rely on the development of hybrid models that combine different modelling approaches, each one designed to model a specific data type or specific modelling problem. In this sense, they might support the development of multicellular and multi-scale frameworks to model complex cellular systems. Examples of such hybrid multicellular approaches are starting to become available. For instance, a multi-scale ABM has been developed to integrate the physical dimension and cell signalling [136], but we anticipate that newer and more powerful hybrid frameworks, which exploit a much more diverse range of modelling approaches and data types, will be become broadly available soon. Yet, coming closer to clinical decision-making, there is very encouraging work combining ODEs and machine learning towards a hybrid system capable of deciding the clinical intervention path: adjuvant vs. neoadjuvant chemotherapy. For instance, the work in [47] combined the nonlinear Gompertzian tumour growth model with neural networks to simultaneously learn the tumour growth curve and the pharmacokinetics of Paclitaxel to predict the therapy course of action for a breast cancer patient. This work emphasises the advantages of hybrid predictive systems combining machine learning and mechanistic modelling in oncology.

For the first time, we have the data, the tools, and the mindset to finally tackle cancer complexity. Exploiting the vast and heterogeneous data from in vitro and in vivo studies along with the multitude of models, we can now build learning systems capable of predicting patient-specific tumour development and, subsequently, patient-tailored treatments. For instance, modelling clonal evolution under antigen receptor CAR T therapies could decipher the activation mechanism of the engineered T cells that describe the dynamical interplay between tumour and immune cells for a certain patient. Subsequently, ABMs could simulate in detail the physics of tumour–host interactions after immune checkpoint therapy and predict the shrinkage or growth depending on the mutational burden of each patient. Equipped with such intimate insights into a patient's tumour evolution, we can consolidate predictive models to simultaneously capture tumour dynamics, therapeutic drug response and kinetics, as well as the therapy outcome.

As one can already see, networks provide an excellent modelling, analysis, and processing substrate to investigate in depth a complex disease like cancer. By describing the relevant interactions between tumour and host before and after therapy and at different scales, we are able to guide the individualised assessment of each tumour and predict the optimal therapeutic intervention for each patient.

**Author Contributions:** C.A., R.B. and M.R.M. designed the perspective structure. M.R.M. structured and wrote Sections 2.1 and 2.2. R.B. structured and wrote Section 2.3. C.A. structured and wrote Section 2.4. C.A., R.B. and M.R.M. wrote and reviewed all other sections. All authors have read and agreed to the published version of the manuscript.

**Funding:** MRM's work was supported by Horizon 2020 (H2020) programs 826121, 765158, 813545, and 955321. R.B. was supported by the Engineering and Physical Sciences Research Council of the United Kingdom (EP/S001433/1).

**Conflicts of Interest:** The authors declare no conflict of interest.

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
