# Peer review of "The Multiple Dimensions of Networks in Cancer: A Perspective"

_symmetry, doi:10.3390/sym13091559_

Round 1

Reviewer 1 Report

The manuscript summarizes the developments in computational oncology tools based on the network theory, a useful approach for modelling and investigating the abnormal cellular processes associated with cancer, and for the simulation of both cancer development and therapy design.

This manuscript represents a good perspective, and it is suitable for being published on this journal.

Nevertheless, I think some revisions are necessary before publication.

  1. Overall, the manuscript is well written, however a language editing is recommended to correct some grammatical errors.
  2. In the Section 1 Network approaches in cancer genomic research some machine learning models’ applications are mentioned, such as the description of the phenotypic stages of a tumor and the suggestion of personalized therapy sequences. The authors should mention other applications such as:
  • to predict the benign or malignant nature of a tumor in a significant subspace of features (Massafra et al. Radiomic Feature Reduction Approach to Predict Breast Cancer by Contrast-Enhanced Spectral Mammography Images);
  • to detect lymph node metastasis in pre-operative stages (Pomarico et al. A Proposal of Quantum-Inspired Machine Learning for Medical Purposes: An Application Case).
  1. In the Section 4 Learning mechanistic interaction networks the authors highlighted the role of machine learning in the development of personalized medicine. For this purpose, an interesting approach that could be cited is illustrated in the following work: Amoroso et al. A Roadmap towards Breast Cancer Therapies Supported by Explainable Artificial Intelligence.

Author Response

We would like to thank the reviewer for the overall feedback and the punctual suggestions to improve the manuscript. We have strengthened Section 2.1 and 2.4.2 with more example applications of tumor phenotypic staging and the use of explainable AI in therapy planning. We have also referenced relevant work (see references 45 and 125, respectively). For updated content tracking please check highlighted text and lines 145 and 467, respectively.

Reviewer 2 Report

The aim of the study was to describe different perspecitves regarding the use of network theory to advance cancer research. Topic is interesting and Authors should be commended for the excellent work. Study is well performed and deal out a specific topic which will change the next future of medical diagnosis process. However, there are just a few minimal aspects thati Authors need to review in order to improve overall quality of manuscript. Background should be improved in order to include applications of radiomics in genitourinary cancers (A Radiomics Nomogram for the Preoperative Prediction of Lymph Node Metastasis in Bladder Cancer - 2017 Nov 15;23(22):6904-6911. doi: 10.1158/1078-0432.CCR-17-1510. Epub 2017 Sep 5.). Actually, the most useful tools to make prediction in clinical practice are the nomograms, which will be replaced by computer networks described by Authors. Therefore this specific aspect chould be discussed (Development and external validation of nomograms predicting disease-free and cancer-specific survival after radical cystectomy - World J Urol. 2015 Oct;33(10):1419-28. doi: 10.1007/s00345-014-1465-4. Epub 2014 Dec 27.). Finally, in the era of quality of life, which is one of the goal for physicians, Authors need to discuss possible implications of Multiple dimensions of network in cancer health related quality of life (Comparison of Patient-reported Health-related Quality of Life Between Open Radical Cystectomy and Robot-assisted Radical Cystectomy with Intracorporeal Urinary Diversion: Interim Analysis of a Randomised Controlled Trial - Eur Urol Focus. 2021 Mar 9;S2405-4569(21)00059-6. doi: 10.1016/j.euf.2021.03.002.).

Author Response

We would like to thank the reviewer for the overall feedback and the punctual suggestions to improve the manuscript. We have strengthened Section 2.4.2 with more example applications of nomograms and their use alone and in combination with learning networks in predictive models of cancer and cancer outcome. We have also referenced relevant work (see references 127 to 132, respectively). For updated content tracking please check highlighted text and lines 474 to 485.

Reviewer 3 Report

In the present article (The Multiple Dimensions of Networks in Cancer: A Perspective), authors present a clinical review of the utility of neural networks in different cancer clinical scenarios. The paper mainly highlight four different situation where neural networks can play a significant role for clinical or treatment response prediction. This is a well-written review which takes us through some of the more important utilities of neural networks in cancer research.

To significantly improve their work, when authors expose the usefulness of neural networks for mechanistic modelling, they should include some references to the ability of neural networks for specifically prediction of cancer metastatic progression. For example, there are some recent works (1) that compare the ability of neural networks with other classical markers, for prediction of metastatic progression.   

(1) She Y, Jin Z, Wu J, Deng J, Zhang L, Su H, Jiang G, Liu H, Xie D, Cao N, Ren Y, Chen C. Development and Validation of a Deep Learning Model for Non-Small Cell Lung Cancer Survival. JAMA Netw Open. 2020 Jun 1;3(6):e205842. doi: 10.1001/jamanetworkopen.2020.5842.

Author Response

We would like to thank the reviewer for the overall feedback and the punctual suggestions to improve the manuscript. We have strengthened Section 2.4.2 with the suggested work (see reference 125). For updated content tracking please check highlighted text and line 468.

Reviewer 4 Report

The paper gives a very interesting perspective on the latest developments in mathematical and computational oncology, with specific attention to network theory.

The paper is well written and sound. I recommend acceptance with the possibility of at most minor adjustments as described in the next paragraph.

If the authors feel it might improve the manuscript further, they might want to consider to explicitly describe some of the models. For example, they can write down specific ODEs of interest and describe them, specific DNNs and describe them, and then put them together in the hybrid approaches. The authors can then conclude what of those models would be interesting to know in future research when they arrive in their "Outlook"-section.

Author Response

We would like to thank the reviewer for the overall feedback and the punctual suggestions to improve the manuscript. Due to constraints in the size of the review, instead of a full exercise that combines ODEs and ANNs, we have pointed to recent work emphasizing the advantages of such an approach in simultaneously learning tumor growth curves (i.e. what ODEs would provide) and pharmacokinetics for clinical decision making (i.e. adjuvant vs. neoadjuvant chemotherapy). This conclusion is now emphasized in the Outlook. For updated content tracking please check highlighted text and lines 598-605.

Reviewer 5 Report

The review is clearly presented, includes a large volume of cited material and should be of a high interest for readers in the areas of cancer and computational biology. I would expect to read slightly more structured insight on a comparison and evaluation of different methods of analysis and interrogation of the networks (when and why different methods are applied?). However, the authors made a clear focus on the complexity of the network-based approach, which is beautifully  illustrated on the example of cancer immunotherapy. No any review is supposed to cover all the aspects of the reviewed area, and I believe that this manuscript can be accepted for a publication in its current version.

Author Response

We would like to thank the reviewer for the overall feedback and the suggestion to improve the manuscript. We have added a more structured insight in the comparison and evaluation of different methods of analysis and use of the networks to support the overall perspective in the “Unifying the dimensions” section. For that, we have extended the examples presented in Figure 8. For updated content tracking please check highlighted text and lines 547 - 565.

Reviewer 6 Report

The review by Axenie et al. is a well written review on the topic of mathematical and machine learning models in oncology. The manuscript only needs minor modification before being appropriate for publication.

Minor comments:

In figure 1. Please add to the figure legends full name of applications (Like MRNET). Also add references, so that the reader can find more relevant information regarding the different applications.

Figure legend 2 and figure 2 is not understandable as an independent figure. Please either make a new figure that illustrates the text where it is referred to, remove the figure as it might be unnecessary, or provide a more detailed figure legend so that it can be understood what it is trying to illustrate without going to the reference.

Miner language changes:

Line 38: “The current study attempts to unify different perspectives regarding the use of network theory to advance cancer research.” Defining this as a study is not correct, better to use review.

Line 63: “…tumor suppressor genes BRCA1 or BRCA2,…”. When referring to genes and RNAs (transcriptome data), use italic.

Line 74: “In then becomes…”Change to “It”.

Line 89: “On the omics level…” Change to “At the...”

Line 112: “pathways, type of interactions, fetc, have also been proposed.” Should it be etc, and not fetc?

Line 121: “…as it will be discussed in Section…” remove “it will be”.

When listing, be consistent with comma use. Either always use before and/or (American English), or never (British English). In most cases, comma has been used before and/or. Example of not used: Line 124: “This formalism results in discrete, deterministic and parameter-free models…”

Line 168: “ex vivo” should be written in itallic

Figure legend 3: “B) Networks within agents can modelled…” Is it missing a “be” before “moddeled”?

Figure legend 4: “Example ABM…” Don’t use abbreviations in figure legend without describing them.

Figure legend 4: “…has important effects healthy as well as cancerous tissue”. Missing “on”.

Line 361: “…learnt from data.” Please use a more specific term than just data.

Figure legend 6: “…tumor cell is an example biological substrate.” Missing “of” after example.

Figure legend 6: “…responsible to recognize the kill malignant cells.” Please rewrite.

Line 409: “MDA-MB-231 [86], MDA-MB-435[87] breast cancer cell lines, …” add and/or between MDA-MB-231 and MDA-MB-435. Or move “breast cancer cell lines” before MDA-MB-231 and MDA-MB-435

Line 423: “…of the breast affected by tumor..” Change to “by the tumor”.

Line 452: Write in-vivo in italic.

Line 469: Write in-silico in italic.

Line 502: “Our study grounds this view…” Change “study” to “review”.

Line 563: in-vitro and in-vivo in italic.

Author Response

We would like to thank the reviewer for the overall feedback and the punctual suggestions on figures and typos that supports us in improving the manuscript. Figure 1 and 2 legends have been updated to include a description and reference to the methods (in Figure 1), as well as to provide the basic background for the reader to understand the figure (Figure 2). For updated content tracking please check highlighted text and figures’ captions.